# Physical Activity Patterns, Circadian Rhythms, and Aggressive and Suicidal Behavior among a Larger Sample of the General Population Aged 15 to 34 Years

**DOI:** 10.3390/jcm12082821

**Published:** 2023-04-12

**Authors:** Habibolah Khazaie, Farid Najafi, Azita Chehri, Afarin Rahimi-Movaghar, Masoumeh Amin-Esmaeili, Mahdi Moradinazar, Ali Zakiei, Yahya Pasdar, Annette Beatrix Brühl, Serge Brand, Dena Sadeghi-Bahmani

**Affiliations:** 1Sleep Disorders Research Center, Kermanshah University of Medical Sciences, Kermanshah 6734667149, Iran; hakhazaie@gmail.com (H.K.); bahmanid@stanford.edu (D.S.-B.); 2Social Development and Health Promotion Research Center, Kermanshah University of Medical Sciences, Kermanshah 6734667149, Iran; faridsn2000@gmail.com; 3Department of Psychology, Kermanshah Branch, Islamic Azad University, Kermanshah 6714673159, Iran; azitachehri@yahoo.com; 4Iranian National Center for Addiction Studies (INCAS), Tehran University of Medical Sciences, Tehran 1419733141, Iranmamines1@jhmi.edu (M.A.-E.); 5Research Center for Environmental Determinants of Health, School of Public Health, Kermanshah University of Medical Sciences, Kermanshah 6734667149, Iranyahya.pasdar@kums.ac.ir (Y.P.); 6Center for Affective, Stress and Sleep Disorders, Psychiatric Clinics of the University of Basel, University of Basel, 4002 Basel, Switzerland; annette.bruehl@upk.ch; 7Division of Sport Science and Psychosocial Health, Department of Sport, Exercise and Health, University of Basel, 4002 Basel, Switzerland; 8Addiction Research Prevention Center, Kermanshah University of Medical Sciences, Kermanshah 6734667149, Iran; 9School of Medicine, Tehran University of Medical Sciences, Tehran 1419733141, Iran; 10Center for Disaster Psychiatry and Disaster Psychology, Psychiatric Clinics of the University of Basel, University of Basel, 4002 Basel, Switzerland; 11Department of Psychology and Department of Epidemiology, Stanford University, Stanford, CA 94305, USA

**Keywords:** aggression, self-injury, suicidal ideation, physical activity patterns, circadian rhythms

## Abstract

Background: From a psychological perspective, aggressive behavior, non-suicidal self-injury and suicidal behavior could be considered dysfunctional coping strategies. Poor sleep patterns may further increase such dysfunctional coping. In contrast, regular physical activity may have the power to counteract such dysfunctional coping. Given this background, the aim of the present study was to combine categories of circadian rhythms as a proxy of normative sleep patterns and categories of physical activity patterns, and to associate these categories with aggressive behavior, non-suicidal self-injury and suicidal behavior among a larger sample of adolescents and young adults, aged 15 to 34 years. Method: A total of 2991 (55.6% females) individuals aged 15 to 34 years of the so-called Ravansar non-communicable disease cohort study (RaNCD) took part in this study. Participants completed self-rating questionnaires covering circadian-related sleep patterns, regular physical activity, socio-demographic information and dimensions of aggression, non-suicidal self-injury and suicidal behavior. Results: In a first step, both sleep patterns (circadian rhythm disorder: yes vs. no) and physical activity patterns (high vs. low) were dichotomized. Next, participants were assigned to one of four prototypical clusters: No circadian sleep disorders and high physical activity (“Hi-Sleep-Hi-PA”); no circadian sleep disorders and low physical activity (“Hi-Sleep-Lo-PA”); circadian sleep disorders and high physical activity (“Lo-Sleep-Hi-PA”); circadian sleep disorders and low physical activity (“Lo-Sleep-Lo-PA”). Projecting these four clusters on dimensions of aggressive behavior, non-suicidal self-injury and suicidal behavior, the following findings were observed: Participants of the “Hi-Sleep-Hi-PA” reported the lowest scores for aggressive behavior, self-injury and suicidal behavior, compared to participants of the “Lo-Sleep-Lo-PA” cluster. No differences for aggressive behavior, self-injury and suicidal behavior were observed among participants of the “Hi-Sleep-Lo-PA” and the “Lo-Sleep-Hi-PA” clusters. Conclusions: It appeared that the combination of favorable circadian sleep patterns and high physical activity patterns was associated with lower aggressive behavior, lower self-injury and suicidal behavior as proxies of favorable psychological functioning. In contrast, persons reporting high circadian sleep disorders and low physical activity patterns appeared to demand particular attention and counseling for both their lifestyle issues (sleep and physical activity) and their dysfunctional coping strategies.

## 1. Introduction

There is extensive evidence that sufficient and restoring sleep is essential for accurate and precise cognitive, emotional, behavioral and interactional processes [1,2,3,4,5]. More specifically, sufficient and restoring sleep is associated with a higher performance of emotion regulation [6,7,8,9,10,11,12], cognitive processes such as a higher performance of the working memory [13,14,15,16,17,18,19,20,21,22,23,24,25] and both working memory and motor learning [26,27,28]. Further, restoring sleep is associated with higher impulse control [29] and with higher self-control [30]. Importantly, sleeping in accord with normative circadian rhythms was also associated with lower impulsivity, that is to say, with higher impulse control [29]. Here, normative circadian rhythms mean sleeping in accord to the nighttime and to current social norms and expectations. Unsurprisingly, non-restoring sleep and circadian rhythm disorders were associated with higher scores for aggressive behavior both among the general population [31,32] and in the forensic context [33].

Further, non-restoring sleep was associated with both a higher risk for non-suicidal self-injury [9,34,35] and suicidal behavior. The latter was observed among individuals with major depressive disorder, specifically [36], and generally among individuals with psychiatric diagnoses [37], among different cohort studies [38] including adolescents [39,40,41,42], and in longitudinal studies [43]. As such, it appears plausible that circadian sleep rhythm disorders as a proxy of poorly adjusted sleep schedules should be associated with higher scores for non-suicidal self-injury and suicidal behavior.

Given this background, we considered these observations and asked whether and to what extent restoring sleep and circadian patterns might be associated with scores for aggressive and suicidal behavior among a larger sample of adolescents and young adults aged 15 to 34 years.

Beside regular and restoring sleep, there is now sufficient evidence that regular physical activity and exercising are associated with a broad variety of beneficial health outcomes. For example, adults self-reporting regular and moderate to high exercising patterns were at lower risk for adverse cardiovascular health outcomes and earlier mortality incidences [44,45]. Further, regular physical activity and exercise were associated with higher odds to deal with stress [46,47,48,49,50,51,52,53,54]. In this view, regular physical activity and exercise were associated with higher scores for self-esteem and self-control among children, adolescents [55] and adults [56,57,58]. As such, it appeared plausible that individuals scoring high on physical activity patterns would also report lower scores for aggressive behavior, non-suicidal self-injury, and suicidal behavior as a proxy of low self-control. Importantly, Baiden et al. [41] showed in their large cross-sectional study with 13,659 adolescents aged 14 to 18 years (51.8% females) that higher scores for physical activity patterns were associated with lower scores for suicidal ideation.

Overall, the current state of the literature suggests that both a circadian rhythm aligned with nighttime and with socially expected patterns, and regular physical activity patterns and exercise should be associated with higher scores for self-control. Here, higher self-control is understood as the umbrella term for low aggressive behavior, low non-suicidal self-injury and low suicidal behavior.

### The Present Study

In the present study, we considered the observations described above and tested whether and to what extent the combination of circadian rhythms as a proxy of favorable sleep patterns and regular physical activity and exercise might be associated with patterns of aggressive behavior, non-suicidal self-injury and suicidal behavior. To this end, we applied a technique successfully employed before [59]. We dichotomized circadian rhythms disorders into the groups “yes” and “no”, and we dichotomized physical activity patterns into the groups “high” and “low”, leading to four prototypical clusters, which were labelled as follows: No circadian sleep disorders and high physical activity (“Hi-Sleep-Hi-PA”); no circadian sleep disorders and low physical activity (“Hi-Sleep-Lo-PA”); circadian sleep disorders and high physical activity (“Lo-Sleep-Hi-PA”); circadian sleep disorders and low physical activity (“Lo-Sleep-Lo-PA”). Next, we projected these clusters on dimensions of aggressive behavior, non-suicidal self-injury and suicidal behavior. In doing so, we sought to identify adolescents and young adults at increased risk for unfavorable behavior based on their sleep and physical activity patterns.

## 2. Methods

### 2.1. Participants

Participants were randomly selected from the so-called Ravansar non-communicable disease cohort study (RaNCD; see [60]). The RaNCD study consists of a total of 10,065 individuals aged 15 to 34 years residing in Ravansar County (Kermanshah Province, northwestern Iran). A subsample was selected as follows: First, of the whole sample, those not exclusively living and residing in Ravansar City or in the catchment area were excluded, leading to a subsample of about 5500 eligible participants. Second, of those, 3000 living and residing for at least six years in Ravansar City or in the catchment area were further randomly selected. Third, eligible participants were fully informed about the study aims, the voluntary participation and the anonymous data handling. Further, all eligible participants self-declared to be physically and mentally healthy. Nine of the selected individuals were excluded from the study process due to incomplete interviews, leading to a total sample of 2991 participants.

### 2.2. Data Collection

Data collection in the first phase of the study took place between April 2015 and April 2017. The data collection site was the Cohort Ravansar Center, where selected individuals visited the center and were interviewed by trained psychologists. Participants were assured that their information would remain confidential; thereafter, they signed the written informed consent. Each interview lasted about an hour, and the quality of the work was under the control of a quality supervisor.

### 2.3. Measures

#### 2.3.1. Socio-Demographic Information and Family History Form

Participants reported the following socio-demographic information: Gender (female = 1; male = 2); age (years); education (no schooling; primary school; secondary school; high school; higher education); civil status (single = 1; married = 2; divorced/widowed = 3); job status (employed = 1; unemployed = 2; student = 3; housewife = 4).

#### 2.3.2. Sleep Patterns and Circadian Rhythms

To assess sleep patterns and circadian rhythms, participants reported on their sleep duration (h), the sleep onset time (min), the number of awakenings after sleep onset (nr), the duration of napping during the day (never, <30 min; 30–120 min; >120 min), and shift work (yes vs. no). Next, following the International Classification of Sleep Disorders (ICSD-2) criteria, circadian rhythm disorders were classified as follows: People who went to bed earlier than 9 pm and woke up earlier than 5 am were assigned to the category of Advanced Sleep Period (Disorder; ASPD); in contrast, people who went to bed later than 2 am and woke up later than 11 am were assigned to the category of Delayed Sleep Period (Disorder; DSPD) [61].

#### 2.3.3. Physical Activity

To assess the degree of physical activity, participants indicated the number of days per week where they were at least moderately physically active for at least 10 min. Moderate physical activity was operationalized as a definitively faster and more intensive breathing frequency and the feeling of sweating, when compared to the default status of rest. This single question was taken from the International Physical Activity Questionnaire (IPAQ) [62].

#### 2.3.4. Suicidal Ideations

To assess suicidal ideations, participants completed the Persian version of Suicidal Thoughts and Behaviors Questionnaire (STBQ). The Persian version of this questionnaire is a modified format used in the Health Mental World (WMH) Survey [63]. This questionnaire includes nine items related to suicidal ideation, serious suicidal thoughts, suicidal plans and behavior, the number of suicidal plans and behavior, number of suicide attempts and the number of referrals to medical centers. In the present study, only information about the suicide plan and the history of suicide attempts (all yes/no) were analyzed. This measure has good inter-rater reliability in Iranian samples [63].

#### 2.3.5. Physical Aggression

To assess physical aggression, participants answered to typical items such as destroying assets, self-injury, domestic violence, violence towards another person and any further aggression within the last 12 months. Answers were dichotomized in yes (=1) or no (=0). This measure has a suitable inter-rater reliability in the general population of Iran (kappa agreement coefficient = 0.6) [63].

### 2.4. Statistical Analysis

All statistical analyzes were performed using SPSS^®^ 28.0 (IBM Corp., Armonk, NY, USA) for Windows. We tested two-sided, and the alpha level of significance was set at *p* < 0.05. Data related to continuous variables were reported as means and standard deviations and categorical variables were reported as frequencies and percentages.

The preliminary calculations provided the classification of nominal and ordinal variables into dichotomous or three-category variables: Advanced Sleep Period Disorder (ASPD); Delayed Sleep Period Disorder (DSPD); shift work: no = 0; yes = 1. Sleep duration categories were merged into three categories: 1–5 h/night; 5–9 h/night; >9 h/night). Daily napping was merged into four categories: Never; <30 min; 30–120 min; >120 min.

Next, a two-stage cluster analysis (TSCA) was performed to identify the clusters. This analysis method was used due to the high sample size and the existence of qualitative classification variables [64]. The TSCA determines the order of importance of the classification variables involved in predicting the model and automatically determines the number of clusters. The fit of the model was determined by Schwarz’s Bayesian information criterion (BIC) using the average silhouette coefficient. Next, and based on the above-mentioned procedures, we followed Brand et al. [59]: We dichotomized circadian rhythms disorders into “yes” and “no”, and we dichotomized physical activity patterns into “high” and “low”, leading to four prototypical clusters, labelled as follows: No circadian sleep disorders and high physical activity (“Hi-Sleep-Hi-PA”); no circadian sleep disorders and low physical activity (“Hi-Sleep-Lo-PA”); circadian sleep disorders and high physical activity (“Lo-Sleep-Hi-PA”); circadian sleep disorders and low physical activity (“Lo-Sleep-Lo-PA”). These four clusters were used to calculate possible differences in distributions related to categorical socio-demographic variables (X^2^-tests). For continuous variables, ANOVAs were performed.

In the last step, a multi-nominal logistic regression analysis was performed to investigate the role of clusters in predicting aggressive behavior and suicide attempts; Cluster 3 (“Lo-Sleep–Hi-PA”) was considered a reference cluster.

## 3. Results

### 3.1. Sleep Patterns, Physical Activity Patterns; Four Prototypical Clusters

Table 1 shows the descriptive and inferential statistical indices of the participants’ sleep-related information and physical activity levels, separated by the four prototypical clusters.

### 3.2. Sleep- and Physical Activity-Related Characteristics of the Four Clusters

Cluster 1 (Hi-Sleep-Hi-PA): Typically, participants belonging to this cluster did report a regular circadian sleep; napping during the day occurred seldom; the total sleep time was predominantly between five to nine hours per night; shift work was rarely reported and hours of physical activity per week were high.

Cluster 2 (Hi-Sleep-Lo-PA): Typically, participants belonging to this cluster did report a regular circadian sleep; napping occurred often, and often for more than 120 min/d; total sleep time was predominantly between five to nine hours per night; shift work was rarely reported and hours of physical activity per week were low.

Cluster 3 (Lo-Sleep-Hi-PA): Typically, participants belonging to this cluster did report some circadian sleep irregularities; napping occurred sometimes; total sleep time was predominantly between five to nine hours per night, while about 11% also reported to sleep more than 9 h/night; shift work was sometimes reported and hours of physical activity per week were moderately high.

Cluster 4 (Lo-Sleep-Lo-PA): Typically, participants belonging to this cluster did report circadian sleep irregularities; napping occurred often; total sleep time was often between five to nine hours per night, while about 22% also reported to sleep more than 9 h/night; shift work was sometimes reported and hours of physical activity per week were low.

### 3.3. Socio-Demographic Information, Separated for the Four Prototypical Clusters

Table 2 reports the descriptive and inferential statistical indices of participants’ socio-demographic information separately for the four prototypical clusters. For gender, relative to Cluster 2 (Hi-Sleep-Lo-PA) and to the total sample, Cluster 1 (Hi-Sleep-Hi-PA), Cluster 3 (Lo-Sleep-Hi-PA) and Cluster 4 (Lo-Sleep-Lo-PA) reported more female than male participants.

For age, compared to all other clusters, Cluster 4 (Lo-Sleep-Lo-PA) reported the highest percentage of participants aged 30–34 years.

For educational levels, clusters classified with high sleep also reported the highest percentages of participants with no schooling.

For civil status, Cluster 4 (“Lo-Sleep-Lo-PA”) reported the lowest percentage of married participants but also the highest percentage of divorced/widowed participants.

For job status, Cluster 4 (“Lo-Sleep-Lo-PA”) reported the highest percentage of students and unemployed people.

### 3.4. Patterns of Aggression, Self-Injury, and Suicidal Ideation between the Four Prototypical Clusters

Table 3 reports the descriptive and inferential statistical indices of aggression, self-injury, and suicidal ideation between the four prototypical clusters.

To run a multi-nominal logistic regression analysis, Cluster 3 (“Lo-Sleep–Hi-PA”) was considered a reference cluster.

For destroying assets, participants with aligned circadian rhythms and high physical activity levels per week (Cluster 1; Hi-Sleep-Hi-PA) reported the statistically significantly lowest risk. In contrast, participants with high circadian rhythm disorders and low physical activity levels per week (Cluster 4; Lo-Sleep-Lo-PA) reported the statistically significantly highest risk for aggressive behavior.

For self-injury, participants with aligned circadian rhythms and irrespective from high or low physical activity patterns (Cluster 1: Hi-Sleep-Hi-PA; Cluster 2: Hi-Sleep-Lo-PA) reported the statistically significantly lowest risks.

For violence towards another person or for aggression as a whole, participants with aligned circadian rhythms and high physical activity levels per week (Cluster 1; Hi-Sleep-Hi-PA) reported the statistically significantly lowest risk.

For suicidal ideation (suicide attempts; lifetime and in the past 12 months), participants with high circadian rhythm disorders and low physical activity levels per week (Cluster 4; Lo-Sleep-Lo-PA) reported the statistically significantly highest risks.

## 4. Discussion

The aim of the present study was to combine categories of circadian rhythms as a proxy of normative sleep patterns and categories of physical activity patterns, and to associate these categories with aggressive behavior, non-suicidal self-injury and suicidal behavior among a larger sample of adolescents and young adults, aged 15 to 34 years. Based on participants’ circadian rhythms and physical activity patterns, four prototypical clusters were identified. As a general observation, participants reporting an aligned circadian sleep and high physical activity patterns (Cluster 1; Hi-Sleep-Hi-PA) were also reporting sufficient sleep per night, a low degree of napping during the day and the lowest risks of aggression towards other people and self-harm as a proxy of dysfunctional coping strategies. In contrast, participants reporting a poorly aligned circadian sleep and low physical activity patterns (Cluster 4; Lo-Sleep-Lo-PA) were also reporting insufficient sleep per night, a high degree of napping during the day and the highest risks of aggression towards other people and self-harm as a proxy of dysfunctional coping strategies. For participants reporting a combination of high or low alignment with circadian rhythms and reporting high or low physical activity patterns (Cluster 2: Hi-Sleep-Lo-PA; Cluster 3: Lo-Sleep-Hi-PA), no specific and statistically significant peculiarities could be identified.

The novelties of the present study were as follows: The combination of a favorable lifestyle of sufficient sleep and physical activity (Cluster 1) was associated with functional coping strategies (namely low aggressive harm to others and to oneself in terms of suicidal ideation), and such a pattern was identified among a larger sample of the general population aged 15 to 34 years, thus covering a broad range of individuals in socio-psychologically distinctive stages ranging from middle adolescence to young adulthood. In contrast, risks of aggressive behavior and suicidal ideation were highest among those reporting a misalignment of circadian rhythms and low weekly physical activity (Cluster 4). Again, the novelty of the present study is that such patterns were observed among a larger sample of the general population ranging from middle adolescence to young adulthood.

In our opinion, the present results mirror previous observations. For the associations between favorable sleep patterns and non-aggressive (self-harm and harm of others), previous studies reported that sufficient and restoring sleep was related to a higher performance of emotion regulation [6,7,8,9,10,11,12], to a higher impulse control [29] and to higher self-control [30]. Importantly, sleeping in accord with normative circadian rhythms was also associated with lower impulsivity, that is to say, with higher impulse control [29]. Unsurprisingly, non-restoring sleep and circadian rhythm disorders were associated with higher scores for aggressive behavior, both among the general population [31,32,65,66,67,68,69] and in the forensic context [33]. Data from the participants assigned to Cluster 4 (Lo-Sleep-Lo-PA) confirmed this. More specifically, non-restoring sleep was associated with both a higher risk for non-suicidal self-injury [9,34,35] and suicidal behavior [70,71,72]. The latter was observed among individuals with major depressive disorder, specifically [36], and generally among individuals with psychiatric diagnoses [37] among different cohort studies [38], including adolescents [39,40] and in longitudinal studies [43]. Further, Baiden et al. [41] showed among a sample of 13,659 14 to 18 years old adolescents that the odds to report suicidal ideations was 1.35-fold higher among those adolescents with insufficient and non-restoring sleep. Importantly, besides insufficient sleep, further predictors for suicidal ideations were female gender, a history of traditional and cyber bulling, feelings of sadness and hopelessness, substance use and being slightly or very overweight. In sum, Baiden et al. [41] evidenced the complexity to understand the associations between suicidal ideation, sleep and psychological functioning. Ahmed et al. [42] further corroborated such complexity: Among a smaller sample of 157 individuals in Saudi Arabia aged 10 to 24 years, a higher odds for repeated suicide attempts was associated with older age, family problems, psychiatric issues and a higher number of hospitalizations.

The novelty of the present result is that the association between poor sleep and suicidal ideation was observed among a larger sample of the general population ranging from mid adolescence to young adulthood, thus embracing two lifetime periods particularly under pressure as individuals take specific, important and serious decisions as regards vocational orientation, mating, independence from the family of origin, dealing with peers, building up a family, securing economic independence and stability or balancing work, leisure time and family, just to name but a few [73].

While the present study confirmed previous results (see above), the quality of the data does not allow a deeper introspection as to why. For want of direct evidence, we rely on previous observations.

First, poor sleep and misaligned circadian rhythms were associated with higher scores for depression, anxiety and bipolar disorders [74,75,76,77,78]. As such, and given that such mental health issues were not assessed in the present study, it is conceivable that dimensions of depression, anxiety and bipolar disorders biased two or more dimensions in the same or opposite directions. Second, shift work was associated with higher scores for suicidal ideation [79]. However, while the prevalence of shift work did statistically differ between the four clusters, the descriptive statistics did not appear to suggest a significant impact of shift work on sleep and behavior (see Table 1). However, participants reporting poor sleep and low physical activity patterns (Cluster 4; Lo-Sleep-Lo-PA) also reported to be unemployed. It appears that unemployment is critical, at least in regard to suicide attempts: Tarolla et al. [80] identified unemployment/underemployment as predictor for another suicide attempt within the four weeks following an initial attempt in adults. Therefore, unemployment should be considered an independent possible risk factor for suicidal behavior.

Second, Yoo et al. [81] experimentally manipulated the sleep duration among healthy participants and observed that such an experimentally induced sleep restriction was associated with a reduced activity of the medial-prefrontal cortex (MPFC); the MPFC is considered key for the inhibitory, top-down control of amygdala function, resulting in contextually appropriate emotional responses to (negative) stimuli. Given this, we assume that individuals in the present study reporting poor sleep had altered MPFC and amygdala functions, which at a behavioral level was mirrored by higher scores for aggression and self-harm.

In regard to physical activity, the present results replicate findings that suggest higher physical activity patterns (though in combination with favorable sleep) were associated with lower scores for aggression and self-harm. Unsurprisingly, among 13,659 adolescents aged 14–18 years, higher physical activity levels were associated with lower scores for suicidal ideation [41]. While again the present data do not present direct evidence of the underlying psychological mechanisms, we claim that the pattern of results might be explained as follows: First, higher scores for physical activity were associated with higher scores for self-esteem and self-control among children, adolescents [55] and adults [56,57,58]. As such, it appeared plausible that individuals scoring high on physical activity patterns did also report lower scores for aggressive behavior, non-suicidal self-injury and suicidal behavior as a proxy of low self-control. Second, higher scores for physical activity were associated with higher scores for coping with stress [46,47,48,49,50,51,52,53,54]. Again, such observations match well with the present pattern among individuals in Cluster 1 (Hi-PA) and Cluster 4 (Lo-PA); importantly, such high or low physical activity patterns were associated with low or high suicidal ideations in previous studies [41,82,83,84].

Interestingly, cluster characteristics suggested that neither favorable sleep alone nor high physical activity patterns alone had the power to be favorably associated with very low scores for dimensions of aggression and self-harm, as observed in the Clusters 2 and 3. Rather, it appeared that that combination of sleep and physical activity patterns conferred to particularly favorable or unfavorable scores for aggression and self-harm. As such, participants reporting a particularly favorable lifestyle (good sleep; high physical activity) were also those reporting low aggression and self-harm, while the opposite was true for those participants reporting a particularly unfavorable lifestyle (poor sleep; low physical activity).

Despite the novelty of the results, the following limitations must be considered. First, participants did not report on symptoms of depression and anxiety, even though such symptoms are associated with poor sleep and poor physical activity patterns. As such, future studies assessing a larger sample of the general population should consider symptoms of depression and anxiety. Second, and relatedly, more recent studies on Cognitive Disengagement Syndrome (CDS; formerly Sluggish cognitive tempo (SCT) [85,86,87,88]) showed that adolescents and young adults scoring high on CDS were also reporting poor sleep [88,89] and higher scores for suicidal ideations [90]. As such, future studies should consider assessing CDS/SCT. Third, dichotomizing variables may bear the risk of reducing the quality of data [91], though dichotomizing variables and performing multi-nominal logistic regressions are common procedures in the field of epidemiology. Fourth, the cross-sectional design does preclude any causality. As such, it is not clear whether sleep and physical activity patterns influenced a person’s coping strategies (namely aggression and suicidal ideation), as the opposite is also conceivable. Future longitudinal designs will help to answer possible causal relationships. Fifth, suicidal ideation does not equate to suicide attempts. As such, a major health issue for both the individual and the public health remain unassessed [42,80]. Future studies in this field should pay more attention to this serious mental health concern.

## 5. Conclusions

Among a larger sample of the general population aged 15 to 34 years, it appeared that participants with poor sleep and low physical activity demand particular attention. As such, participants seemed to be at increased risk for aggressive behavior, both against other people and against oneself. Further cross-sectional and longitudinal studies could corroborate the results of the present study.

## Figures and Tables

**Table 1 jcm-12-02821-t001:** Descriptive and inferential statistical indices of participants’ sleep-related information and physical activity levels, separated by the four prototypical clusters.

			Cluster 1	Cluster 2	Cluster 3	Cluster 4	X^2^	*p*
			Hi-Sleep-Hi-PA	Hi-Sleep-Lo-PA	Lo-Sleep-Hi-PA	Lo-Sleep-Lo-PA		
N			610	1997	270	114		
ASPD (yes)			0	0	4 (1.5)	0	1.33	0.013
DSPD (yes)			0	0	8 (3)	114 (100)	92.10	0.001
Daily sleep (min)	no		341 (56)	812 (40.7)	13 (4.8)	9 (7.9)	1007.67	0.001
	<30		165 (27.1)	871 (43.6)	37 (13.7)	3 (2.6)	292.29	0.001
	30–120		80 (13.1)	207 (10.4)	103 (38.1)	22 (19.3)	1584.77	
	>120		23 (3.8)	107 (5.4)	117 (43.3)	80 (70.2)	169.78	
Shift work (yes)			59 (9.7)	90 (4.5)	31 (11.5)	12 (10.5)	72.29	0.001
Total sleep time (h)		1–5	43 (7.1)	108 (5.4)	18 (6.7)	0	59.23	0.001
		5–9	518 (85.1)	1686 (84.4)	223 (82.6)	88 (77.2)	2524.19	
		>9 h	48 (7.9)	203 (10.2)	29 (10.7)	26 (22.8)	308.56	
Physical activity (h/week) (M ± SD)			6.81 ± 0.47	0.79 ± 1.13	2.99 ± 0.89	0.29 ± 0.57	F	0.001
							8756.13	

ASDP = advanced sleep period disorder; DSPD = delayed sleep period disorder.

**Table 2 jcm-12-02821-t002:** Socio-demographic information, separated by the four prototypical clusters.

Characteristics		Total	Cluster 1	Cluster 2	Cluster 3	Cluster 4	X^2^	*p*
	Cluster Characteristics	Total Sample	Hi-Sleep-Hi-PA	Hi-Sleep-Lo-PA	Lo-Sleep-Hi-PA	Lo-Sleep-Lo-PA		
	N	2991	610	1997	270	114		
		N (%)	N (%)	N (%)	N (%)	N (%)		
Sex	Female	1663 (55.60)	284 (46.50)	1209 (60.50)	116 (43.00)	54 (47.40)	60.90	0.001
	Male	1328 (44.40)	326 (53.50)	788 (39.50)	154 (57.00)	60 (52.60)		
Age	15–19	313 (10.50)	69 (11.30)	194 (9.70)	29 (10.70)	21 (18.40)	44.11	0.001
	20–24	610 (20.40)	123 (20.20)	389 (19.50)	59 (21.90)	39 (34.20)		
	25–29	902 (30.20)	175 (28.70)	619 (31.00)	69 (25.60)	39 (34.20)		
	30–34	1166 (39.00)	242 (39.70)	795 (39.80)	113 (41.90)	15 (13.20)		
Education	Illiterate	11 (0.40)	0	10 (0.50)	0	1 (0.90)	44.77	0.001
	Primary school	481 (16.10)	71 (11.70)	362 (18.10)	42 (15.60)	6 (5.30)		
	Middle school	530 (17.70)	92 (15.10)	371 (18.600)	49 (18.10)	18 (15.80)		
	High school	1222 (40.90)	264 (43.30)	795 (39.8)	113 (41.90)	49 (43.00)		
	Higher education	747 (25.00)	182 (29.90)	459 (23.00)	66 (24.400)	40 (35.10)		
Civil status	Single	1227 (41.00)	266 (43.70)	761 (38.10)	124 (45.9)	75 (65.80)	59.94	0.001
	Married	1668 (55.80)	326 (53.50)	1167 (58.40)	140 (51.90)	33 (28.90)		
	Divorced/widowed	96 (3.20)	17 (2.80)	69 (3.50)	6 (2.30)	6 (5.30)		
Job status	Employed	1098 (36.70)	258 (42.40)	694 (34.80)	122 (45.20)	24 (21.10)	91.60	0.001
	Unemployed	280 (9.40)	56 (9.20)	174 (8.70)	26 (9.60)	24 (21.10)		
	Student	324 (10.80)	72 (11.80)	191 (9.60)	30 (11.10)	31 (27.20)		
	Housewife	1289 (43.10)	223 (36.60)	938 (47.00)	92 (34.10)	35 (30.70)		

Notes: Hi-Sleep-Hi-PA = High sleep quality and high physical activity patterns; Hi-Sleep-Lo-PA = High sleep quality and low physical activity patterns; Lo-Sleep-Hi-PA = Low sleep quality and high physical activity patterns; Lo-Sleep-Lo-PA = Low sleep quality and low physical activity patterns.

**Table 3 jcm-12-02821-t003:** Prevalence of aggression, self-injury and suicidal ideation reported separately for the four clusters.

		Cluster 1	Cluster 2	Cluster 3	Cluster 4	*p* Value (X^2^)
		Hi-Sleep-Hi-PA	Hi-Sleep-Lo-PA	Lo-Sleep-Hi-PA	Lo-Sleep-Lo-PA	
N		610	1997	270	114	
Destroying assets	N (%)	152 (25)	399 (20)	76 (28.1)	48 (42.1)	0.001 (453.21)
	OR(CI)	0.63 (0.48–0.85)	0.85 (0.61–1.17)	1	1.85 (1.17–2.92)	
Self- self-injury	N (%)	49 (8)	162 (8.1)	34 (12.6)	23 (20.2)	0.001 (184.69)
	OR(CI)	0.61 (0.38–0.96)	0.61 (0.41–0.91)	1	1.75 (0.98–3.14)	
Domestic violence	N (%)	173 (28.4)	655 (32.8)	84 (31.1)	30 (26.3)	0.001 (1040.63)
	OR(CI)	0.88 (0.64–1.20)	1.08 (0.82–1.42)	1	0.79 (0.48–1.29)	
Violence towards another person	N (%)	55 (9)	119 (6)	31 (11.5)	16 (14)	0.001 (112.09)
	OR(CI)	0.49 (0.32–0.74)	0.77 (0.48–1.22)	1	1.26 (0.66–2.40)	
Any aggression	N (%)	291 (47.8)	935 (46.8)	144 (53.3)	68 (59.6)	0.016 (1299.87)
	OR(CI)	0.77 (0.60–0.99)	0.81 (0.60–1.07)	1	1.29 (0.83–2.02)	
Suicide attempt in lifetime	N (%)	23 (3.8)	106 (5.3)	13 (4.8)	14 (12.3)	0.003 (155.03)
	OR(CI)	0.78 (0.39–1.56)	1.11 (0.61–2)	1	2.77 (1.26–6.09)	
Suicide attempt in past 12 month	N (%)	7 (1.1)	30 (1.5)	3 (1.1)	5 (4.4)	0.019 (42.38)
	OR(CI)	1.02 (0.26–3.99)	1.36 (0.41–4.48)	1	4.28 (1–18.26)	

## Data Availability

Data are made available to experts in the field; such experts should clearly state their hypotheses and describe and declare, where and how data (or even parts of them) are securely stored, and above all not shared with third parties.

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
