# Peer review of "Physical Activity Patterns, Circadian Rhythms, and Aggressive and Suicidal Behavior among a Larger Sample of the General Population Aged 15 to 34 Years"

_jcm, 2023, doi:10.3390/jcm12082821_

Round 1

Reviewer 1 Report

I read with interest the present articol by Habibolah Khazaie et al. on Physical activity patterns, circadian rhythms, and aggressive and suicidal behavior among a larger sample of the general population.

The aim of the article is interesting; the authors wanted to examine the relationship between circadian rhythms, physical activity patterns and aggressive behavior both non-suicidal  self-injury and suicidal behavior among a larger sample of adolescents and young adults.

1.      In the introduction the authors describe the characteristics of the subjects showing a high risk of suicidal behaviour. Several study explored these characteristics releved a significant correlations between some dimension and suicidal behavior.

I suggest some references: DOI: 10.1016/j.psychres.2019.112579, DOI: 10.1708/1794.19532, DOI: 10.2147/RMHP.S245175

2.      In the methods section the inclusion/esclusion criteria and the method of recruiting patients results unclear. I suggest to include a flow chart to better describe the recruting process.

3.      In the conclusion I would suggest they to emphasize the need to reply their findings in future studies.

Author Response

We thank Reviewer #1 for their valuable and encouraging comments, which helped us to improve the quality of the manuscript. Please find the revised manuscript and the detailed point-by-point-response attached as separate files. Again, we thank Reviewer #1 for the care devoted to the present manuscript.

Reviewer 2 Report

The present study explored the potential association of physical activity patterns, circadian rhythms, and aggressive and suicidal behavior among a larger sample of the general population in Iran. This study is novel in intention and appropriate in methodology, especially the very large sample size of this study allowed for a very detailed examination of the effects of physical activity patterns and circadian rhythms on aggressive and suicidal behavior in various subgroups, which also led to very solid conclusions.

I have two suggestions for this article, and I hope the author will give an appropriate response.

First, the populations included in this study were actually all adolescents and young adults, rather than an all-age population, so the title of this study is suggested to be appropriately modified to highlight the characteristics of the populations included in this study.

Second, in the table3 section of the results, which I think is the most critical part of the results of this study, what is the author's rationale for choosing Lo-Sleep-Hi-PA as the control group? Why was Hi-Sleep-Lo-PA not chosen as the control group?

Author Response

We thank Reviewer #2 for their valuable and encouraging comments, which helped us to improve the quality of the manuscript. Please find the revised manuscript and the detailed point-by-point-response attached as separate files. Again, we thank Reviewer #1 for the care devoted to the present manuscript.
